# Immune-deficient bacteria serve as gateways to genetic exchange and microbial evolution

Wendy Figueroa [1,2,7], Akshay Sabnis[1,2,7], Rodrigo Ibarra-Chávez [3,4], Jamie Gorzynski [5], J. Ross Fitzgerald [5] & José R. Penadés [1,2,6] ✉

Horizontal gene transfer plays a key role in bacterial evolution, yet its efficiency under natural conditions, especially between genetically distinct strains, remains unclear. Using *Staphylococcus aureus* as a model, we found that gene transfer via various mechanisms is significantly restricted between strains from different clonal complexes (CCs), with the notable exception of lateral transduction, which occurs at high frequency. Interestingly, some strains exhibited a promiscuous ability to accept diverse mobile genetic elements. These strains were defective in key immune defences, specifically the Type I restriction-modification systems that normally protect against foreign DNA. A broader analysis revealed that such immune-deficient mutants are widespread within *S. aureus* populations. Our study uncovered a trade-off that may account for their persistence in nature: although these mutants are more susceptible to phage attack, they gain an evolutionary advantage by acquiring new genes - such as those conferring antibiotic resistance - which would enhance survival under selective pressure. These immune-deficient cells act as gateways for foreign DNA, which, once integrated and advantageous, can spread within the same CC. Our findings highlight the role of immune-deficient bacteria in facilitating the emergence of novel virulence factors and antibiotic resistance, emphasising their importance in shaping bacterial evolution.

Horizontal gene transfer (HGT) between bacterial species plays a pivotal role in bacterial evolution, promoting adaptation to diverse environments, the spread of antibiotic resistance, and the emergence of new pathogens. Understanding these mechanisms is crucial for developing strategies to address the antibiotic resistance crisis and mitigate the impact of bacterial pathogens on human health and the environment.

There are three main mechanisms of HGT in bacteria: transformation, conjugation and transduction. During transformation, bacteria take up free DNA fragments from their environment, typically originating from neighbouring lysed cells[1,2]. This DNA can be integrated into the chromosome through RecA-mediated homologous recombination (for chromosomal DNA) or maintained episomally as plasmids. For transformation to occur bacteria must be naturally competent, a trait observed in a limited number of bacterial species[3,4]. Conjugation involves direct cell-to-cell contact where a donor bacterium transfers DNA to a recipient bacterium via a specialised structure known as a type IV secretion system or pilus[5,6]. In transduction bacteriophages (phages) or phage satellites package and transfer bacterial DNA during their lytic cycle. Instead of packaging their own genome they may inadvertently package chromosomal or plasmid DNA from the host, which is then delivered to a new host cell[7]. As with

[1]Department of Infectious Disease, Imperial College London, London, UK. [2]Centre for Bacterial Resistance Biology, Imperial College London, London, UK. [3]Section of Microbiology, Department of Biology, University of Copenhagen, Copenhagen, Denmark. [4]Center for Evolutionary Hologenomics, Globe Institute, University of Copenhagen, Copenhagen, Denmark. [5]The Roslin Institute, Royal (Dick) School of Veterinary Studies, The University of Edinburgh, Easter Bush Campus, Edinburgh, UK. [6]Veterinary School, Universidad CEU Cardenal Herrera, Alfara del Patriarca, Spain. [7]These authors contributed equally: Wendy Figueroa, Akshay Sabnis. ✉e-mail: j.penades@imperial.ac.uk

transformation the transferred chromosomal DNA must undergo recombination in the recipient strain to be incorporated into its genome. Importantly since phages may encode virulence and other essential accessory genes, it is worth noting that phages can also increase the repertoire of genetic information in recipient cells through lysogenic conversion[8]. This process involves the incorporation of phage DNA into the bacterial genome during the lysogenic cycle of a temperate bacteriophage resulting in the phage DNA integrating into the bacterial chromosome and becoming a prophage. Each of these mechanisms of HGT facilitates the exchange of genetic material driving bacterial evolution and the dissemination of traits, such as antibiotic resistance or virulence factors that can contribute to pathogenicity.

The importance of these processes is underscored by the fact that many different mobile genetic elements (MGEs) can be mobilised via various mechanisms. For instance not only MGEs that encode a complete conjugative machinery (such as conjugative plasmids and integrative-conjugative elements) employ conjugation as a mechanism for mobility. Instead other elements such as mobilisable plasmids and integrative-mobilisable elements hijack the conjugation machinery of other co-residing MGEs to facilitate their efficient transfer in nature[9–11].

A similar scenario occurs with phage satellites and their helper phages[12]. While these elements cannot produce all the components required to generate infectious particles they hijack those from their helper phages to promote their preferential packaging and widespread dissemination in nature.

In recent years an important body of work has revealed the molecular basis of these mechanisms. These studies have individually quantified the efficiency of transfer associated with different mechanisms across various bacterial models. However many of these studies suffer from two significant limitations. First the efficiency of transfer of the different mechanisms was analysed individually, making it difficult in many cases to compare the effectiveness of different gene transfer mechanisms against one another. Second in most of these studies the donor and recipient cells used to measure the efficiency of specific systems were genetically identical, neglecting the impact that different bacterial immune systems may have on controlling the flow of genetic information[13–15].

In a recent study we aimed to address the first question in the relevant human and animal pathogen *Staphylococcus aureus* by simultaneously comparing the efficiency of transfer for the most important mechanisms present in this bacterium (conjugation and transduction). It is important to note that *S. aureus* is not naturally competent. Remarkably our results revealed that, contrary to established dogma, which assumed that the mobility of classical genetic elements (via conjugation, transformation or transduction) was higher than that of the bacterial chromosome (via generalised transduction (GT)), the chromosome in this bacterium is more mobile than classical MGEs when transferred via lateral transduction[16]. This finding challenges the conventional understanding of mobility in bacterial genetics. However as previously mentioned, both donor and recipient strains in this study were genetically identical, raising the question of whether these surprising results hold true when different clinical strains are used as donors or recipients in mobility experiments.

Here we initially aimed to address the second limitation by analysing the efficiency of transfer for different mechanisms when unrelated *S. aureus* strains were used as recipients. Our results highlight lateral transduction as the most powerful mechanism of transfer in this species. They also reveal that while the transfer of many MGEs occurs when donor and recipient are isogenic this transfer is blocked when the donor and recipient are genetically unrelated, a process that severely limits the evolution of the recipient cells. Importantly we found that immune-deficient strains naturally occur in nature, acting as gateways for the acquisition of new genetic information. When the immune-deficient bacteria acquire new genes they can then be shared with other members of the community, driving bacterial evolution and the emergence of novel virulence clones.

## Results

### Impact of clonal complexes on horizontal gene transfer and genetic exchange

Like many important pathogens *Staphylococcus aureus* is divided into clonal complexes (CCs), which are groups of genetically related strains descending from a common ancestor. While all strains belong to the same species those from different CCs can be genetically distinct, with differences significant enough in some cases to influence host tropism[17]. When compared *S. aureus* strains belonging to different CCs not only exhibit variations in virulence genes but also possess different immune mechanisms that may significantly influence genetic flow between strains of different CCs. Among the immune systems present in *S. aureus* type I restriction-modification (RM) systems play a crucial role in shaping the genetic structure of this species[18].

To overcome the limitations of a previous study in which we compared different modes of gene transfer in *S. aureus* using donor and recipient strains belonging to CC8[16] we now evaluated the transfer of various MGEs, as well as portions of the bacterial chromosome, using a collection of 27 strains that belong to 8 different CCs. Note that while most of these CCs are human-specific some include strains infecting other hosts (see Supplementary Table 1 for details). For some of the CCs we included multiple strains, allowing us to evaluate the potential existence of differences even among strains within the same CC.

In our first experiment we included the following MGEs: phage 80α[19], SaPIbov1[20] and plasmids pI258[21] and pGO1[22], as well as a chromosomal marker mobilised by 80α by either generalised or lateral transduction[23]. For phage 80α we used a derivative carrying an *ermC* marker, conferring resistance to erythromycin, which allowed us to measure lysogenic conversion. SaPIbov1 is one of the prototypical members of the phage-inducible chromosomal islands (PICIs)[24,25], which are phage satellites that utilise helper phages for mobility. In these experiments we used a SaPIbov1 derivative carrying a *tetM* marker in the *tst* gene[20]. SaPIbov1 employs phage 80α as a helper and in our experiments, we have successfully generated lysates that exclusively contain SaPIbov1 but not 80α, by using an 80α derivative mutant in the phage small terminase subunit[26]. Importantly the SaPIbov1 particles are composed of phage proteins[27], meaning that SaPIbov1 and 80α particles have identical potential to inject the packaged DNA into the recipient cells (identical host tropism). The same phage (80α) was used to generate transducing particles capable of mobilising a chromosomal marker (Cd) via lateral transduction (LT) or plasmid pI258 via GT. Finally the conjugative plasmid pOG1 was used to measure inter-CC conjugation.

The results measuring 80α lysogenic conversion (integration in the bacterial chromosome) were dramatic and unexpected: it essentially did not occur when recipients belonged to a different CC than the donor (CC8) used to generate the phage lysate. The exceptions were two strains belonging to CC1 and one belonging to CC130. Even in these limited cases lysogenic conversion occurred with frequencies that were more than 100 times reduced compared to those observed when strains belonging to CC8 were used as recipients (Fig. 1).

Somewhat less dramatic but still quite impactful were the unexpected results obtained when analysing the transfer of SaPIbov1 or pI258 (via GT). For these MGEs most of the recipient strains were refractory to gene transfer, and as seen with 80α lysogenic conversion, the results obtained in those strains able to acquire foreign DNA were never at the level observed when the donor and recipient belonged to the same CC. Importantly most of the strains capable of acquiring SaPIbov1 were also able to accept pI258, and vice versa, suggesting that the same mechanisms limit the transfer of these elements.

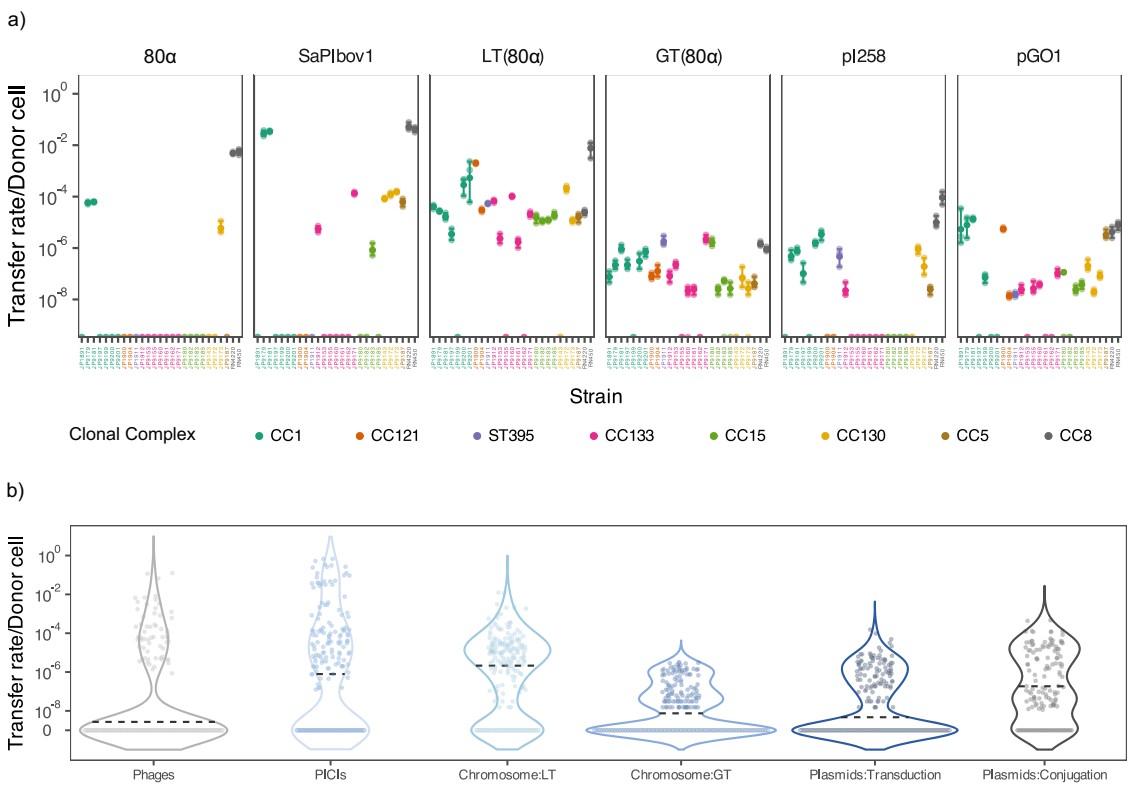

**Fig. 1 | HGT in clinical *S. aureus* strains. a** Transfer efficiency of a selection of MGEs. The x-axis shows the strain of the collection and the y-axis indicates the transfer rate in the particular bacterial isolate ($n = 3$ biological replicates; data are presented as mean values ± SD). The individual data points can be seen in the plot along with error bars representing the standard deviation. One representative of each mobile genetic element (MGE) type is shown and matches the labels of (**b**) (i.e. 80α is a phage representative SaPIbov1 a PICI, LT(80α) a chromosomal marker mobilised by lateral transduction, GT(80α) a chromosomal marker mobilised by generalised transduction, pI258 is a representative plasmid mobilised by generalised transduction, and pGO1 a plasmid mobilised by conjugation). Colours represent strains belonging to the same clonal complex. **b** Violin plots summarising the

transfer efficiency of all MGEs. Each violin represents a different MGE type: the first violin summarises the efficiency of the phages 80α, φ11 and φSLT; the violin corresponding to PICIs includes SaPIbov1, SaPIbov5 and SaPIpT1028; chromosome: LT shows the efficiency of chromosomal markers mobilised by lateral transduction mediated by 80α, φ11 and SaPIbov1; chromosome: GT includes same markers as chromosome: LT but mobilised by generalised transduction; plasmids: transduction includes pI258, pC221 and pUR3912; and plasmids: conjugation shows the efficiency of pGO1 and pC221 mobilised by pGO1. Individual data points are shown as dots whereas the dotted line denotes the median of each group. Individual transfer efficiencies can be found in Supplementary Fig. 3.

In contrast to the limited transfer observed for 80α SaPIbov1, or pI258, the conjugative transfer of pGO1, the 80α-mediated transfer of the chromosomal marker via GT, and especially the transfer of the chromosomal marker via LT occurred in most of the strains used as recipients, with LT offering the highest frequencies of transfer overall. Interestingly while LT and GT shared many of the strains that could accept DNA, there were discrepancies between the strains that were refractory to conjugation versus transduction (Fig. 1, Supplementary Figs. 1 and 2), suggesting the existence of specific mechanisms in these strains that block different processes.

### Lateral transduction is the most efficient mechanism of gene transfer in *S. aureus*

Since the previous study included only a few MGEs and markers located in specific regions of the bacterial chromosome we expanded our repertoire of elements to determine whether the results could be generalised. Specifically we included two additional temperate phages, φ11[28] and φSLT[29], two additional SaPIs (SaPIbov5[30] and SaPIpT1028[15,31]), and two additional chromosomal markers, which can be mobilised by phage LT using φ11[23] or by SaPI LT using SaPIbov1[32]. We also included two additional plasmids pC221[33] and pUR3912[34], which can be mobilised by GT (both) or by conjugation (pC221) with the help of the conjugative plasmid pGO1[33], and an additional conjugative plasmid, pWBG731[33,35]. As summarised in Supplementary Table 2 most of the

elements selected in this study are clinically relevant, encoding virulence or antibiotic resistance genes, as well as immune systems.

The results for all these MGEs and chromosomal markers individually represented for each element in Supplementary Fig. 3 and summarised in Fig. 1b, indicate that lateral transduction transfer (LT), followed by GT and plasmid transfer via conjugation, are the most important mechanisms of transfer in *S. aureus*. In this species however phage, SaPI, and plasmid transfer via GT is severely blocked when strains from different clonal complexes (CCs) are used as recipients.

Finally since all the strains used as recipients were sequenced, we analysed the possibility that the interference observed in the transfer of the MGEs could be related to the presence of similar elements in the recipient strains. Apart from three cases where the strains JP9199 JP9171 and JP9185 had prophages with the same repressor as φSLT, we did not find resident MGEs with the same repressor as the ones used to test mobility, nor did we find any correlation between the presence of MGEs with the same integrase in the strains and the transfer patterns (Supplementary Table 3). Moreover as previously mentioned, since phage-mediated GT and LT worked well, but not the SaPI or plasmid transfer mediated by the same phages—implying that these elements could be injected—the results suggested that the arsenal of immune systems present in the recipient strains determined their ability to block or accept foreign DNA.

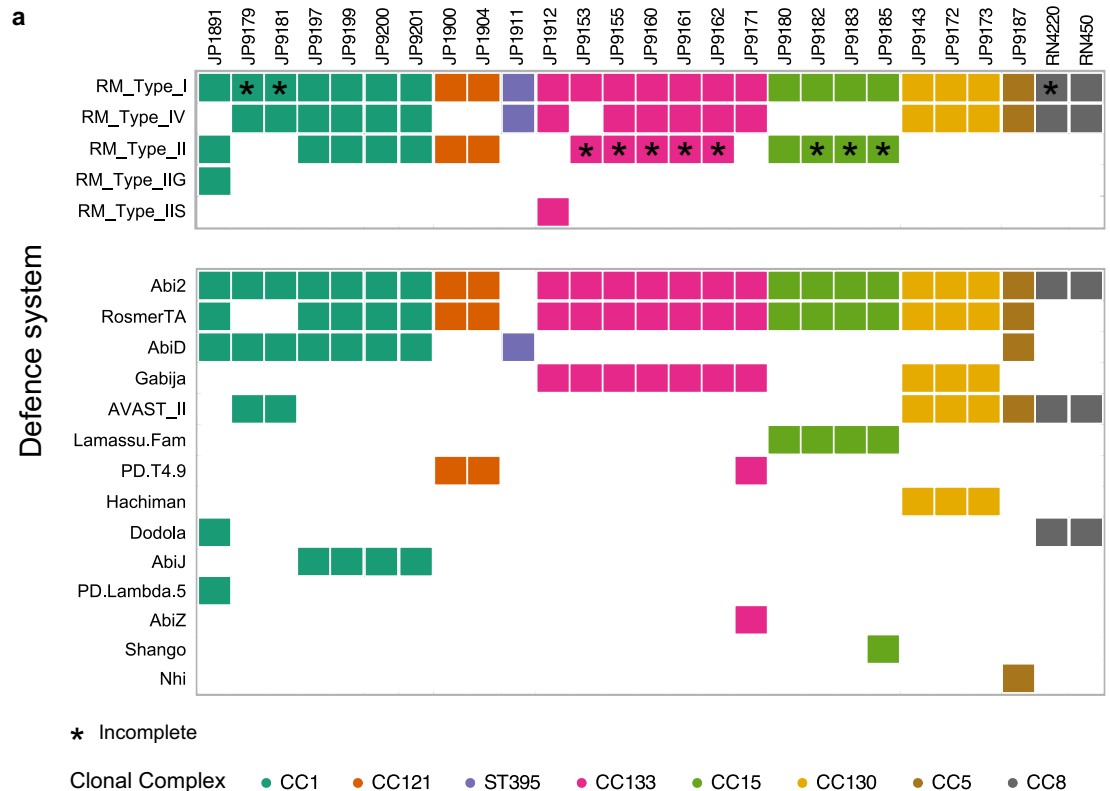

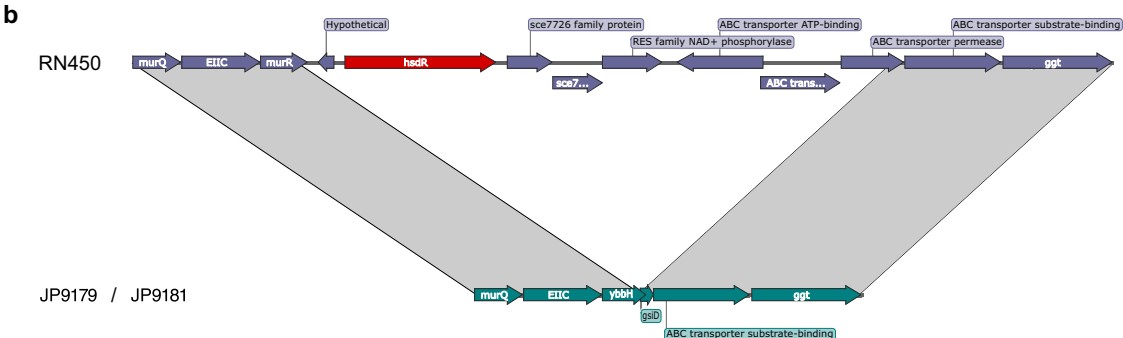

**Fig. 2 | Defence systems present in the *S. aureus* collection. a** Presence of restriction modification (RM) systems (types I, IV, II, IIG and IIS) in our collection (top panel) and other known immune systems (bottom panel). Asterisks indicate that the system in the specific strain is not functional. Colours represent strains belonging to the same clonal complex. **b** Comparison between the strain RN450 that encodes a functional RM system and the type I RM-negative strains JP9179 and JP9181. The panel shows a large deletion involving amongst other genes, the RM restriction subunit *hsdR* (shown in red).

## Promiscuous recipient strains are mutants in type I RM systems

To test this idea further we focused on two strains, JP9179 and JP9181, belonging to CC1, which were able to accept the transfer of all the MGEs and chromosomal markers used in this study. Since three other strains belonging to CC1 were quite refractory to the acquisition of some MGEs the simplest explanation is that JP9179 and JP9181 were immune-deficient, meaning they were defective in bacterial immune systems. To test this possibility we analysed the presence of defence systems in all strains used in these experiments as recipients. The repertoire of defence systems varied between the strains with type I and type IV RM systems, Abi2, and RosmerTA being the most prevalent, present in 100%, 70.3%, 96.2% and 81.4% of the strains, respectively (Fig. 2a).

Related to strains JP9179 and JP9181 which carry type I and type IV RM systems, Abi2, AbiD and AVAST_II (Fig. 2a), we realised that the restriction subunit of the type I RM system present in these strains is defective (Fig. 2b), probably explaining the capacity of these strains to acquire exogenous DNA. In *S. aureus* type I RM systems involve three *hsd* genes (*hsdR*, *hsdM* and *hsdS*) regulating DNA restriction, modification, and sequence specificity, respectively[14]. *S. aureus* usually has two genomic locations encoding the *hsdMS* alleles which cluster with pathogenicity islands vSaα and vSaβ, respectively[15], while *S. aureus* strains only encode one *hsdR* gene, which localises elsewhere in the bacterial chromosome. Previous studies have shown that type I RM variants are associated with specific CCs impacting the frequency of inter-CC HGT[18,36–38] (Supplementary Table 4). Regardless of the

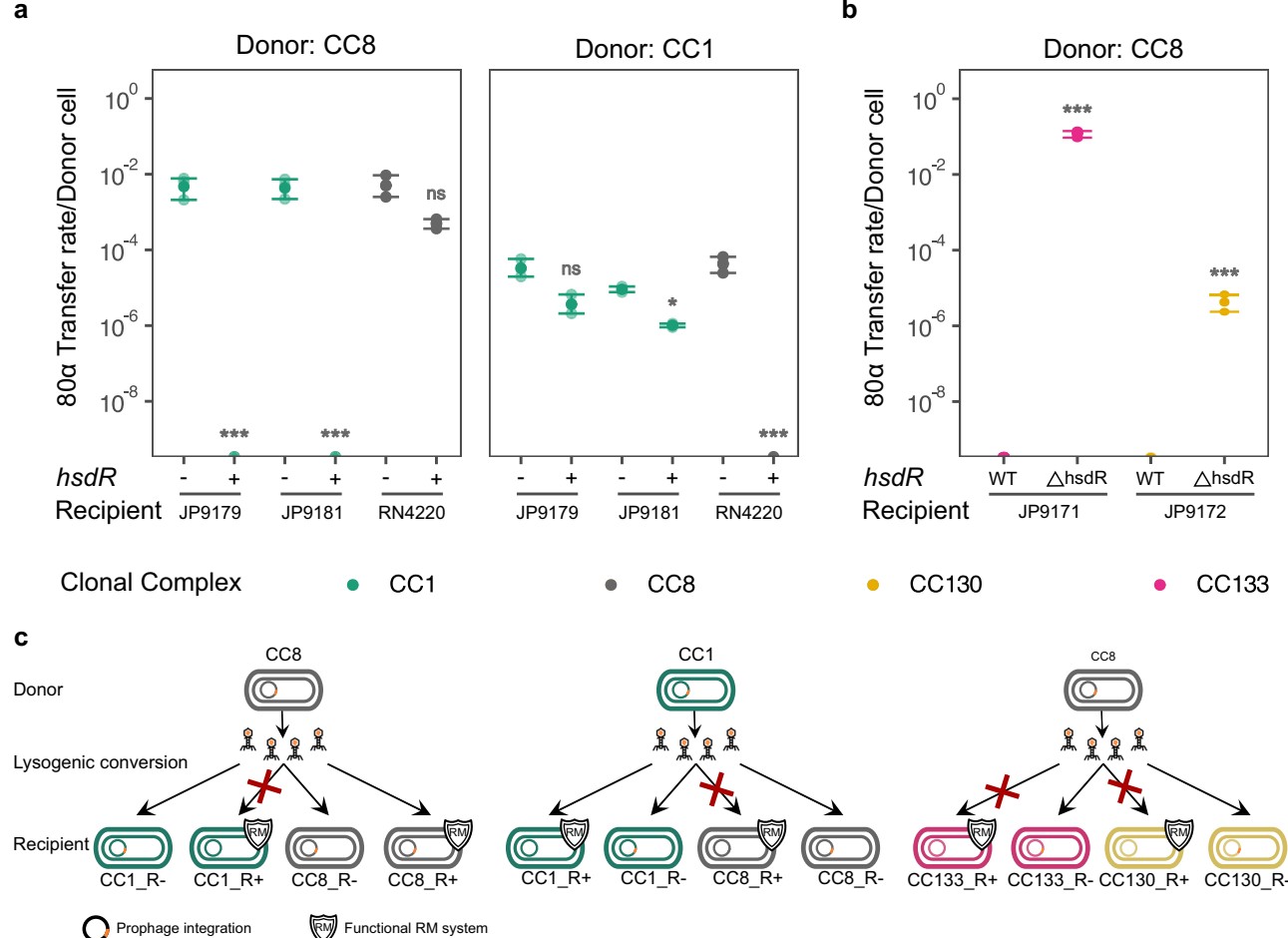

**Fig. 3 | Role of HsdR in MGE transfer. a** Lysogenisation of phage 80α in strains carrying either the wild-type or mutant variant of *hsdR* (*n* = 3 biological replicates; data are presented as mean values ± SD). Natural restriction-deficient (R⁻) strains JP9179 and JP9181, as well as RN4220 with an empty vector (−) or expressing *hsdR* (+), were used as recipients. The left-hand panel shows phage transfer using RN4220 (CC8) as the donor strain. The right-hand panel shows transfer rates using JP9179 (CC1) as the donor. **b** Deletion of *hsdR* in the immuno-competent JP9171

(CC133) and JP9172 (CC130) allows lysogenisation of 80α. In these experiments (*n* = 3 biological replicates; data are presented as mean values ± SD), the donor strain for phage 80α belonged to CC8. **c** Schematic representation of methodology used in (**a**, **b**) experiments. Welch t-tests (two-sided) were used to compare differences in efficiency of transfer between mutants and parental strains in both panels. Asterisks denote the *p*-values (\**p* < 0.05; \*\**p* < 0.01 and \*\*\**p* < 0.001). Colours indicate the clonal complex of the recipient strain.

variants carried in the different strains all of them can interact with the conserved R subunit present in this bacterium. Importantly in both JP9179 and JP9181 strains, there was a deletion in the region containing the *hsdR* gene (Fig. 2b), making these strains mutants in the restriction subunit. However the two copies of the *hsdMS* genes present in these strains are potentially functional, indicating that these strains were unable to digest the incoming DNA, but once inside the cells, they were able to methylate it correctly. This process would favour the transfer of the elements to other immune-competent strains belonging to the same CC.

To confirm that the ability of the JP9179 and JP9181 strains (CC1) to acquire exogenous DNA was a consequence of their defect in the *hsdR* gene we introduced either an empty plasmid or a derivative expressing the *hsdR* gene into these strains and analysed their ability to generate 80α lysogens using phage lysates obtained after induction of the 80α prophage present in a strain belonging to CC8. In these experiments we also introduced these plasmids into RN4220, which belongs to CC8 and is mutant in *hsdR*. Supporting our hypothesis expression of *hsdR* in JP9179 and JP9181 completely eliminated their ability to generate lysogens, while it had no effect on strain RN4220 (Fig. 3a), confirming that JP9179 and JP9181 were immune-deficient due to the absence of

*hsdR*. We then repeated the same experiments using an 80α lysate obtained after induction of the prophage from a CC1 strain. In this case lysogenisation occurred at the same rate in all the CC1 strains but was abolished in the RN4220 strain expressing *hsdR* (Fig. 3a).

To generalise our results we next deleted the *hsdR* gene in strains JP9171 (CC133) and JP9172 (CC130) and tested lysogenic conversion of phage 80α (obtained from a strain belonging to CC8). Supporting the important role of h*sdR* in controlling HGT deletion of this gene in the two recipient strains allowed them to acquire the 80α phage at high frequencies (Fig. 3b).

Finally we tested the idea that once introduced into an immune-deficient cell, the acquired MGEs could then be mobilised intra-CC to other strains, as the immune-deficient cells would be able to methylate the DNA to be recognised as belonging to the CC. To do this we used two 80α lysates: one obtained from RN4220 (CC8) and the other obtained after induction of the prophage present in the immune-deficient strain JP9179 (CC1). The lysates were then used to infect JP9179 JP9181 and JP9197, all belonging to CC1. Note that JP9179 and JP9181 are immune-deficient while JP9197 is immune-competent. While the 80α phage from the CC8 strain was only able to generate lysogens in the immune-deficient cells the one from the CC1 donor was able to

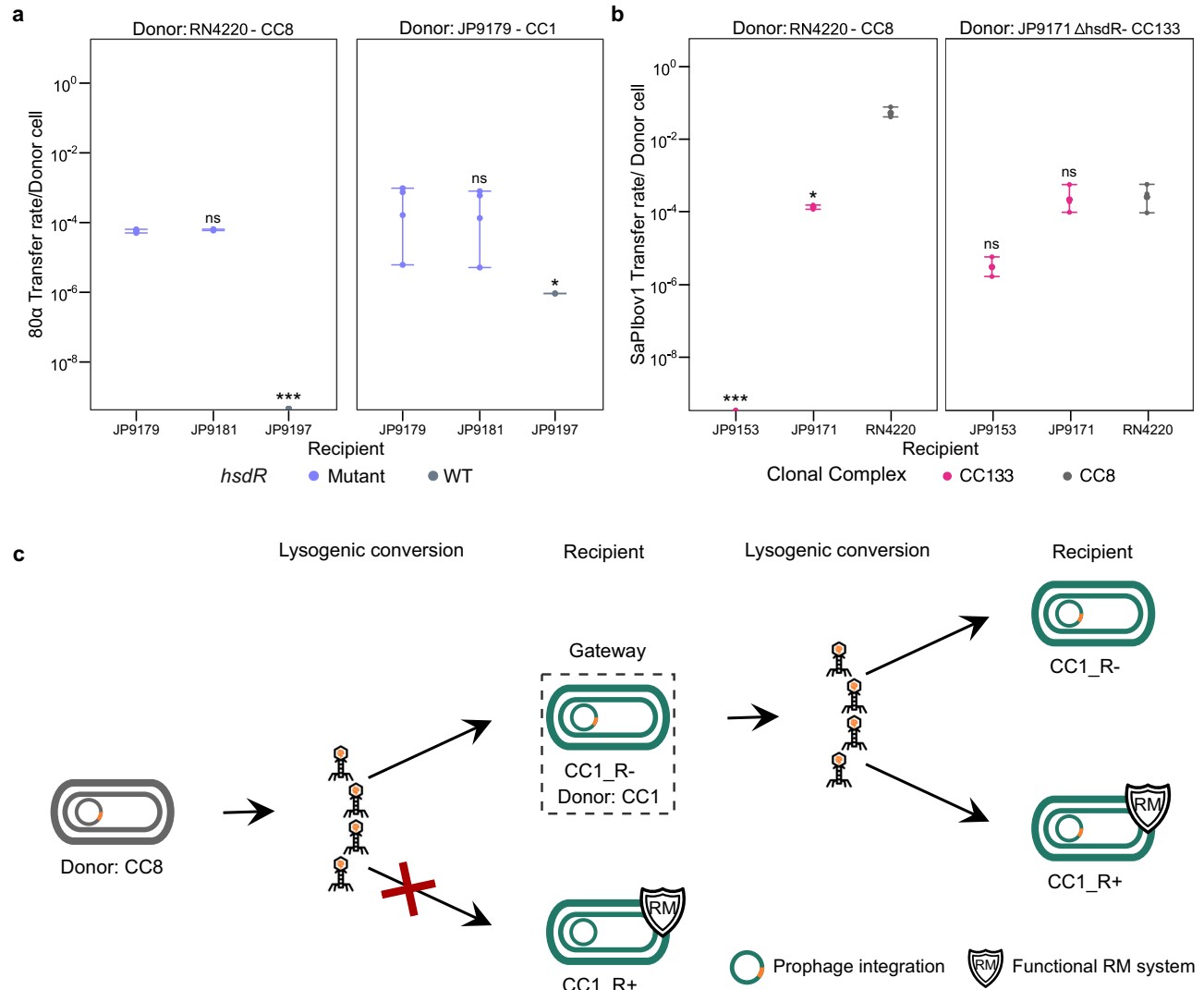

**Fig. 4 | HGT mediated by *hsdR* mutants as a gateway mechanism. a** The left panel shows the transfer efficiency of phage 80α from RN4220 (CC8) into three CC1 strains: JP9179 and JP9181 (natural *hsdR*-deficient mutants), and JP9197 (carrying a functional R subunit). The right panel shows transfer using JP9179 (R-deficient, CC1) as the donor onto the same recipient strains. Colours indicate *hsdR* status: mutant (purple) or wild type (grey). Note that JP9179 and JP9197 even though their names are similar, are completely different strains (see Supplementary Table 5−core SNP distance matrix) with JP9179 being R-deficient and JP9197 R-proficient. **b** The panel shows the transfer efficiency of SaPIbov1 into two CC133 strains (JP9153 that blocks SaPIbov1 transfer, and JP9171, shown in pink) and RN4220 (dark grey), using either RN4220 (CC8) or the type I RM mutant JP9171ΔhsdR (CC133) as donors. Colours indicate the clonal complex of the recipient strain. **c** Schematic representation of methodology used in (**a**, **b**) experiments. Welch t-tests (two-sided) were used to compare differences in efficiency of transfer between mutants and parental strains in both panels. Asterisks denote the *p*-values (\**p* < 0.05; \*\**p* < 0.01 and \*\*\**p* < 0.001).

generate lysogens in all the CC1 strains used as recipients (Fig. 4a). This result confirms that immune-deficient cells that are restriction-deficient but methylation-proficient act as gates for the entrance of foreign DNA which is then shared with the rest of the strains belonging to the same CC. Additionally to test whether this phenomenon was restricted to phages or represented a more general mechanism, we used strains JP9153 and JP9171, both from clonal complex CC133. We had observed that although both strains appeared to carry a functional type I RM system (Fig. 2a), JP9153, but not JP9171, was refractory to the transfer of SaPIbov1 when the island originated from a CC8 strain (RN4220; Figs. 1a and 4b). By contrast when the SaPIbov1 lysate was generated in a derivative of JP9171 carrying an *hsdR* mutation (within CC133), both JP9153 and JP9171 were able to accept the DNA (Fig. 4b). This confirmed that appropriate methylation of the DNA enabled SaPIbov1 transfer to the previously resistant strain when donor and recipient belonged to the same clonal complex (Fig. 4b). The fact that

SaPIbov1 could be mobilised from RN4220 (CC8) to JP9171 but not to JP9153 (both CC133), suggests that the type I RM system in JP9171 is either non-functional or not expressed. Notably JP9171 harbours three additional immune systems (RM_Type_IV, PD.74.9, and AbiZ), which are absent in JP9153 (see Fig. 2a). Whether the presence of these systems interferes with the expression of the type I RM system is currently under investigation.

## hsdR mutants are prevalent in natural populations

The previous results raised an immediate question: how frequently are mutations in *hsdR* found in natural populations? For this 'gateway' mechanism to facilitate HGT effectively these mutations should be present in nature. To test this we analysed all the *S. aureus* complete genomes sequenced strains available in the NCBI database and quantified the number of strains carrying loss-of-function mutations (e.g. stop codon gained and frameshift mutations) in the *hsdR* gene. As a

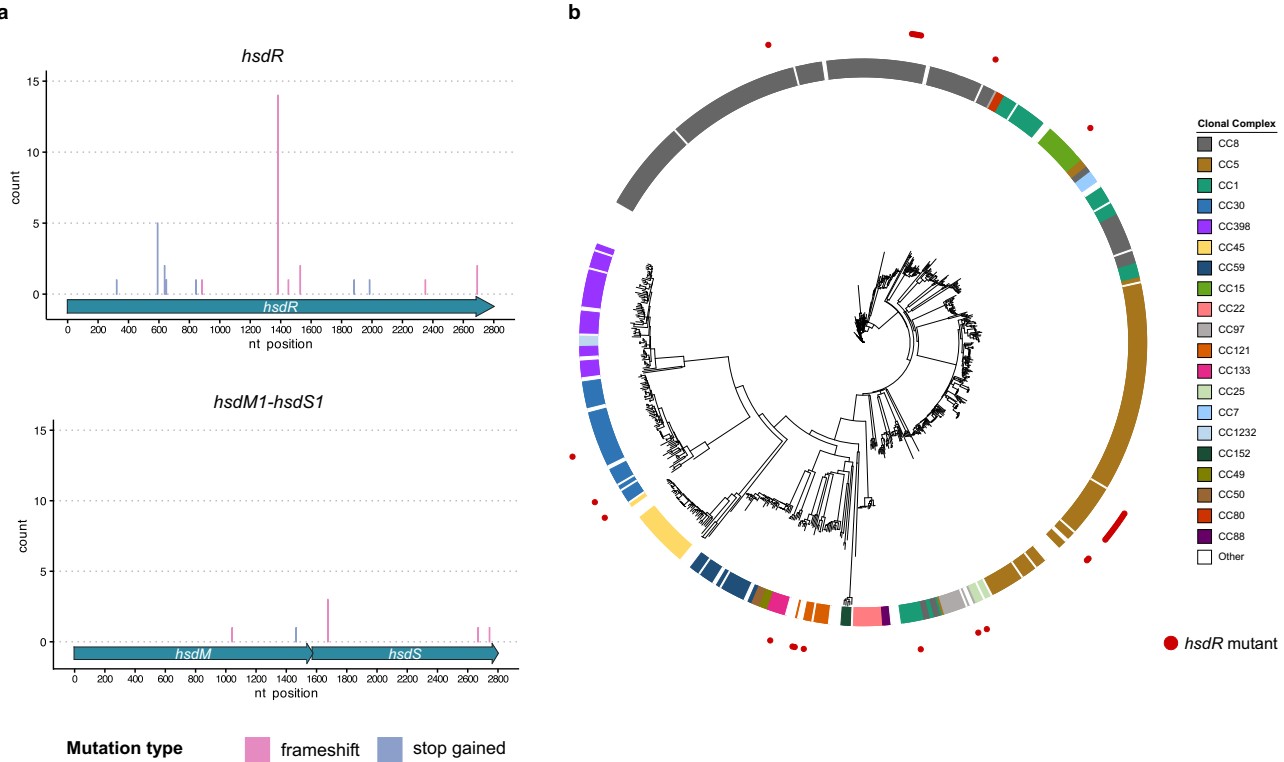

**Fig. 5 | Mutational profile of *hsdR* and neighbouring genes. a** Distribution of loss-of-function mutations ((stop-gained (lilac) and frameshift (pink)) in the type I RM system across all complete *S. aureus* genomes available in GenBank. Mutations in *hsdR* are shown in the top panel while mutations in *hsdM* and *hsdS* are shown in the bottom panels. The x-axis represents the nucleotide position within each gene and the y-axis indicates the number of mutations identified at each position. **b** Mash distance tree of all complete *S. aureus* genomes. Coloured blocks indicate the clonal complex of each isolate and red dots mark strains carrying mutations in *hsdR*.

control for this analysis we also quantified the number of mutations found in the *hsdM* or *hsdS* genes within these strains. Remarkably while we found that 4% of the strains carried loss-of-function mutations in the *hsdR* gene (Fig. 5a and Supplementary Fig. 4), only 0.2% and 0.9% of the strains carried such mutations in either the *hsdM* or *hsdS* genes, respectively (Fig. 5a). Importantly the strains carrying mutations in *hsdR* belonged to different CCs, confirming that the presence of these immune-deficient cells is common across all lineages of *S. aureus* (Fig. 5b).

### Trade-off between phage susceptibility and DNA acquisition drives expansion of *hsdR* mutants

The high prevalence of *hsdR* mutants suggested either the existence of a selective pressure that promotes the expansion of *hsdR* mutants. We therefore speculated that an interesting trade-off may occur in *hsdR* mutants: although they are more sensitive to phage attack which would typically eliminate them from the population, if the acquired DNA provides a significant advantage, these mutants could expand rapidly, potentially explaining their high prevalence in the population.

To test this we mixed JP9179 (CC1), which is an *hsdR* mutant, and JP9197 (CC1), that carries a functional R subunit, and infected the culture with a lysate containing phage 80α (carrying the *ermC* marker) produced by the strain RN4220 (CC8). After infection the population was split into two: one part received no additional treatment, while the other was treated with erythromycin (whose resistance is encoded in the 80α prophage). After 0 4 and 8 h, the number of JP9179 and JP9197 cells present in both populations was measured. Supporting the idea of a trade-off between infectivity and benefits in the absence of the antibiotic, the relative abundance of the immune-deficient cells (JP9179) decreased. However after treatment with the antibiotic, the situation was completely reversed, with the immune-deficient strain

JP9179 dominating the culture (Fig. 6). In conclusion *hsdR* mutants persist in bacterial populations possibly due to a trade-off between phage vulnerability and the acquisition of beneficial foreign DNA, which promotes their expansion under selective pressure.

## Discussion

Previous studies have found that *S. aureus* follows a clonal population structure[39] in which the gene exchange between different lineages seems to be extremely uncommon[40]. One of the main reasons for this is the presence of bacterial defence systems in particular RM systems. RM systems can be found in over 95% of bacterial genomes with the type I RM system present in nearly half of the prokaryotic genomes[41–43]. Waldron et al. have found that nearly all *S. aureus* isolates encode type I RM systems[18]. Interestingly the DNA motifs recognised by this system are unique to each clonal complex[18], and therefore each clonal complex produces a unique DNA methylation pattern. Thus different studies have shown that the distribution of MGEs, such as plasmids and bacteriophages, is dependent on the clonal complex[44,45].

In this study we provide experimental evidence that the transfer of MGEs between different strains is considerably impacted by the lineage to which donor and recipient cells belong. Importantly we demonstrate that this is, surprisingly, not the case for chromosomal genes, which can be transferred efficiently between cells, regardless of the clonal nature of the strains. In particular we confirm that the recently described phage- and SaPI-mediated lateral transduction are by far the most efficient way of HGT in *S. aureus*[23,32]. While in a previous study where we used donor and recipient cells belonging to the same CC, we suggested that this was the case when comparing different mechanisms of transfer[16], the results obtained here, analysing transfer between CCs, are more striking, highlighting LT as the main

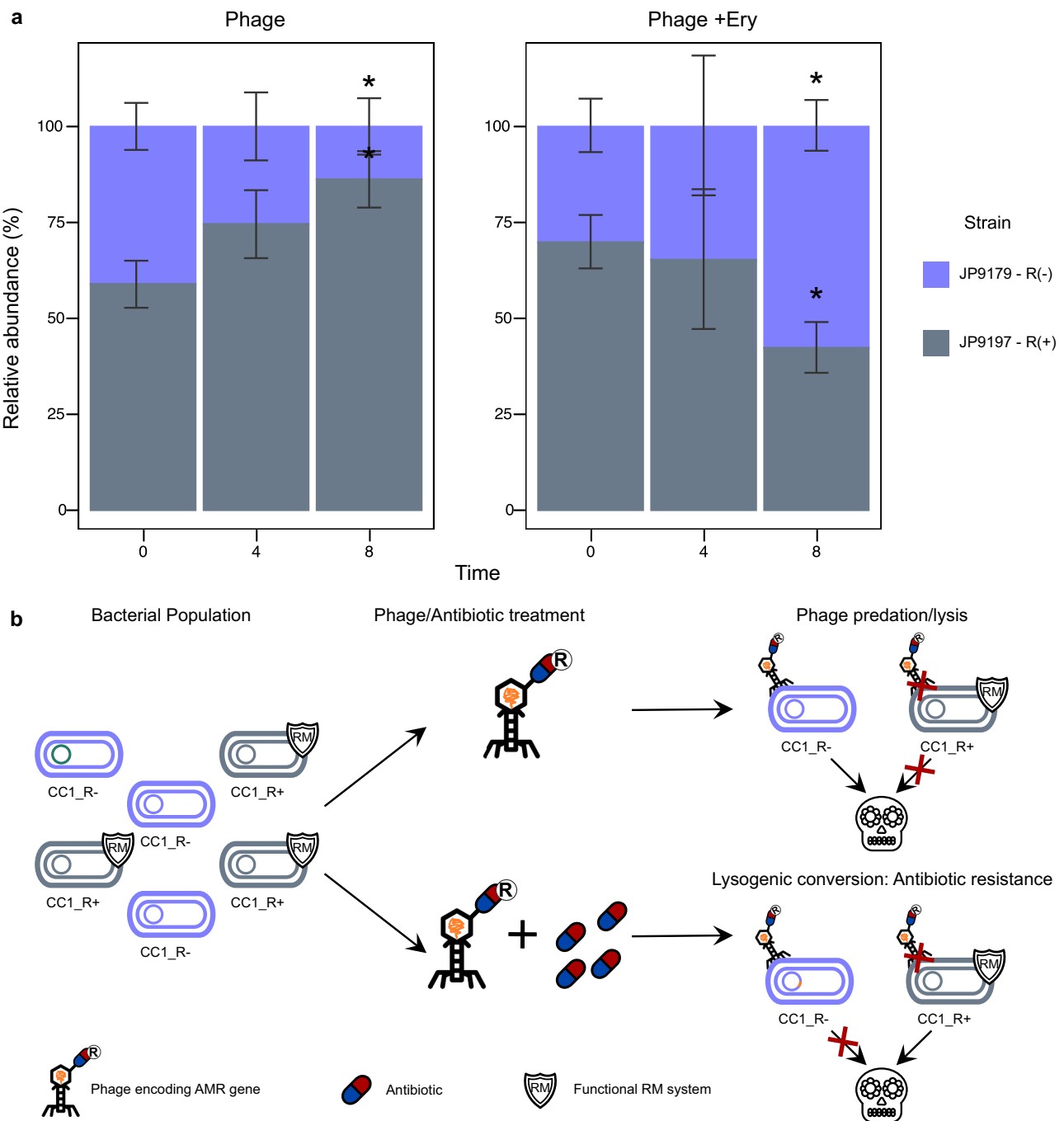

**Fig. 6 | Fitness trade-off of *hsdR* mutants. a** Relative abundance of *hsdR*-proficient (JP9197, shown in grey) and *hsdR*-deficient (JP9179, shown in purple) strains from the same clonal complex grown in co-culture. Cultures were challenged with a phage lysate obtained from the strain RN4220 (CC8), with and without an antibiotic (erythromycin/Ery) to which the phage confers resistance. The plot shows the relative abundance of each strain at 0 4 and 8 h after phage and antibiotic addition. Asterisks indicate statistical differences between the initial proportions and the rest of the time points using an Anova test (*$p < 0.05$). **b** Schematic representation of methodology used in (**a**) experiments.

mechanism of gene transfer in this species. Whether this also occurs in other species is currently under investigation.

Even though here we only experimentally assessed particular MGEs which represents a limitation of our study, our results highlight the role of GT and conjugation as important mechanisms of HGT in *S. aureus*. While GT was previously assumed to occur at very low transfer rates our results indicate that it occurs at frequencies higher than those observed for the transfer of different MGEs, consistently— meaning that it occurs at similar frequencies regardless of the CC used as the recipient. A hypothesis explaining why LT and GT are not

affected by RM systems is provided below. Similarly conjugation appears to occur at a lower frequency than LT but at a higher frequency than GT. The minor role that RM systems may play here could be explained by either the expression of different systems by some conjugative elements that block RM system activity[46] or because conjugation involves the transfer of single-stranded DNA, which is not cleaved by RM systems.

The results obtained with LT and GT raise the question of why if the methylation patterns are the same, are chromosomal genes transferred but not MGEs? There are different ways in which DNA can

be incorporated into the chromosome, such as RecA-mediated homologous recombination and integrase-mediated site-specific recombination. The latter is the mechanism employed by bacteriophages phage satellites, and other MGEs. Specifically MGE-encoded integrases recognise a specific bacterial attachment site (*attB*) and mediate the recombination between this site and the attachment site present in the MGE[47], resulting in the integration of the phage. Since the MGEs must be in a circular form for their integration cleavage by the RM systems results in an MGE that cannot integrate or replicate, even if there is only one recognition site in the whole MGE sequence. On the contrary RecA-mediated integration, which is used by LT or GT (for the mobility of chromosomal genes), can occur with both linear and circular DNA as long as there is sufficient homology between the incoming DNA and the target DNA. Thus even if the incoming chromosomal DNA is cleaved in many pieces, the resulting fragments can still recombine with the chromosome of the recipient cell if still DNA is provided for the double recombination, leading to DNA integration. On this note it has been suggested that SaPIs and plasmids could be more tolerant than phages to cleavage by exonucleases due to the redundancy of their genomes and their recombination capabilities[48]. This could explain why these elements were less blocked than phages but more than chromosomal DNA in our experiments.

The presence of natural mutants defective in the type I RM system represents a complex evolutionary trade-off. On one hand R-deficient strains are more susceptible to phage predation and therefore should be negatively selected. On the other hand having a defective RM system gives them the advantage of being able to horizontally acquire potentially beneficial genes. However there is another layer of complexity to this system: R-deficient mutants can act as gateways for HGT between different clonal complexes. As discussed before the population structure of *S. aureus* means that different clonal complexes rarely exchange MGEs because of the DNA motifs recognised by them. However RM(−) gateways provide the platform to exchange genes between CCs by acquiring, methylating and transferring them to strains from the same clonal complex that were resistant to HGT from other CCs. A similar mechanism was recently observed by Sheng et al., who reported the prevalence of insertion sequences (IS) in defence systems such as CRISPR-Cas, RM and Dnd systems[49]. They found that out of 60 different immune systems 12 carried insertions of IS1 and IS10, likely inactivating them.

The variation in HGT permissiveness observed in our study reflects a broader ecological phenomenon where specific amplifier strains can disproportionately drive the movement of genetic material within a community. As demonstrated by Dionisio et al.,[50] the presence of even a small subpopulation of efficient donors can accelerate the spread of MGEs by several orders of magnitude, preventing the genetic stagnation of the population. In the context of *S. aureus* our data suggest that R-deficient mutants serve this exact role, acting as high-efficiency 'hubs' for inter-clonal exchange. However maintaining such permissiveness comes with an evolutionary downside: the loss of protection against phage predation. This tension between the need for adaptive genetic acquisition and the cost of viral susceptibility has been explored in CRISPR-Cas systems by Jiang et al.,[51] who noted that immune-deficient variants often persist at low frequencies to allow for the intake of beneficial plasmids. Furthermore the epidemiological stability of these strains likely depends on the specific force of infection from beneficial versus deleterious genetic elements, a dynamic modelled by Gandon and Vale[52]. By identifying the R − M+ phenotype as a molecular gateway, our work provides a mechanistic basis for how S. aureus balances these conflicting selective pressures to maintain a flexible yet protected pangenome.

One might assume that R-deficient mutants would carry more MGEs than strains with functional RM systems. However we noted that, interestingly, this was not the case (Supplementary Table 3). We hypothesise that the acquisition of new MGEs in an immune-deficient

strain could result in conflicts amongst them if, for example, they use the host-bacterium replication machinery, or carry defence systems that target one another[15]. Additionally it is possible that there are physical and metabolic constraints that hinder the increasing acquisition of several elements[53]. A recent large-scale study showed that plasmid size and copy number depend on the cell's metabolic capacity with plasmid load consistently accounting for a similar percentage of chromosomal DNA across hosts[54]. Similar constraints may govern the balance between different MGEs explaining the lack of high MGE numbers in immune-deficient strains. Besides these limitations the fact that R-deficient strains are considerably more susceptible to phage predation makes them particularly vulnerable, and therefore, it is likely that spontaneous R-defective mutants are negatively selected in bacterial populations when there is no fitness advantage conferred by the MGE.

Our mutational analysis discovered that mutations in the R subunit are common in *S. aureus* but not in the M or S subunits. Typically RM systems are composed of two enzymatic activities: a methyltransferase (MS) and a restriction endonuclease (R). The R subunit cleaves the invading DNA while the methyltransferase activity allows to differentiate self and incoming DNA by methylating specific DNA motifs[55]. Beyond their function as defence systems it has been reported that methylation by RM systems may play other roles, such as DNA repair or regulation of gene expression[56,57]. The R-deficient isolates in our collection (JP9179 and JP9181) differed by 214 core SNPs (Supplementary Tables 5 and 6), indicating that these strains have a common ancestor[58,59], but have been evolving independently while maintaining a non-functional type I RM system. Therefore it is possible that the selective pressure to have a functional methylation machinery is higher than to have a functional immune system.

Given that we have shown that R-deficient strains act as a gateway for inter-CC HGT it is plausible that other mechanisms are in place to modulate intra-CC genetic exchange. Ultimately the long-term persistence of a lineage depends on a functional equilibrium; excessive HGT may lead to lethal levels of phage predation, while overly restrictive immunity may result in an inability to adapt to changing environments. While strains from the same clonal complex had the same type I RM systems we found additional immune systems that were present in some but not all strains of the same CC (Fig. 2a). For instance in CC1, the type IV and II RM systems, RosmerTA, AVAST II, Dodola, AbiJ and PD-λ−5 were only present in a proportion of the strains. Similarly the type IIS RM system, PD-T4-9, and AbiZ were only found in some strains of the CC133, and Shango was present in only one strain of the CC15, suggesting that these immune systems may play a role in controlling intra-CC gene flow.

The observation that RM systems limit HGT in *S. aureus* is consistent with historical practices in bacterial strain identification. The widespread use of phage typing to distinguish *S. aureus* strains implicitly relied on the fact that certain phages could infect only specific lineages suggesting a barrier to phage-mediated gene transfer across strain boundaries[60]. Although the molecular basis for this specificity was not fully understood at the time it is now evident that RM systems play a central role in restricting the flow of genetic material between strains. This evolutionary phenomenon likely contributed to the genomic stability of individual lineages and helps explain the lineage-specific distribution of MGEs observed in modern *S. aureus* populations[18,36−38]. In recent complementary studies we have demonstrated that LT can mediate the transfer of RM systems located in genomic islands vSaα and vSaβ, leading to strains attenuated in their capacity to exchange genetic information horizontally (Kuang et al., In press, 2026). Further we have discovered that, within CCs, genetic subpopulations have diverged by changes to their complement of RM system variants, suggesting a role in the emergence of new lineages[61].

In summary our study demonstrates that lateral transduction is the most powerful route for HGT in *S. aureus*, with MGE transfer

restricted between strains from different clonal complexes. Since HGT is essential for bacterial evolution we show that bacteria bypass the limitations imposed by the T1RM system by using immune-deficient mutants that are restriction-deficient but methylation-proficient, which are prevalent in nature and serve as gateways, facilitating HGT between different clonal complexes. These findings highlight the role of immune-deficient bacteria in pathogen adaptation and evolution.

## Methods

### Bacterial strains
The recipient strains were kindly gifted by Prof Luca G. Guardabassi. The full list of bacterial strains and MGEs used in this study is provided in Supplementary Tables 1 and 2. Unless otherwise stated bacterial cultures were grown in tryptic soy broth (TSB) at 37 °C with shaking at 120 rpm. Where required the media was supplemented with the following concentrations of antibiotic for MGE/chromosomal marker selection: erythromycin (10 µg/mL), tetracycline (3 µg/mL), cadmium chloride (300 µM), gentamicin (10 µg/mL) and chloramphenicol (10 µg/mL).

### Inductions
Bacterial cultures were grown in TSB at 37 °C shaking at 120 rpm, to an O.D. of 0.15. To induce the prophage(s) and make a phage lysate mitomycin C was added to a final concentration of 2 µg/mL. Induced cultures were incubated at 30 °C for 3 h at 80 rpm. After this time the cultures were left overnight at room temperature, statically. Finally the clear lysates were filtered using a 0.2 µm syringe filter.

### Transductions
Bacterial cultures were grown in TSB to an O.D. of 1.4. For the transduction experiments 100 µL of lysates were added to each culture, along with 5 mM of CaCl$_2$. The cultures were incubated statically at 37 °C for 30 min and then plated onto TSA plates containing 17 mM sodium citrate and the respective antibiotic for the selection of the different MGE. Plates were incubated for 48 h and colonies were counted. The transfer efficiency or transfer event (TE) per donor cell was calculated as the transductant units per mL divided by the donor colony-forming units (CFU) per mL ($6.5 \times 10^7$ CFU/mL for a culture with an O.D. of 0.15).

### Conjugations
Conjugation assays were performed using mixed cultures of the donor strain (RN4220 pGO1-GmR, RN4220 pGO1-GmR + pC221 or RN4220-pWGB731) and the recipients (CdR or EryR mutants of the strains from the collection). Overnight cultures of donor and recipient cells were washed with PBS to remove antibiotics and resuspended in TSB without antibiotics. Equal numbers of donor and recipient cells ($5 \times 10^7$ CFU) were combined. The entire mixture was spotted onto a sterile 0.45 µm pore nitrocellulose filter (MF-Millipore™ Membrane Filter) placed onto TSA. The mixture was allowed to dry onto the filter before incubating at 37 °C for 24 h. Donor- and recipient-only controls were included in parallel. Following incubation filters were transferred to microtubes and vortexed vigorously in 1 ml phosphate-buffered saline to detach the bacteria from the filters for quantification. To determine plasmid transfer frequencies donors and recipients were titrated on TSA and TSA containing the appropriate selective antibiotic. Transfer efficiency was calculated as TEs per donor cell (i.e. number of recipient cells harbouring the plasmids/number of donor cells).

### Defence systems detection
A bacterial immune system search was performed using the software PADLOC and DefenseFinder[62,63]. Additionally RM systems were identified using the REBASE database[43]. The database was downloaded and a BLAST search was performed against the genomes of the collection. The output files from the different programs were compiled together

(Fig. 2a). Plotting of the prevalence of such systems was performed in R 3.4.

### Variants detection
All the complete genomes of *S. aureus* were downloaded from GenBank and were set to start from the gene *dnaA* using the program circlator[64]. We then deduplicated the genomes using the software Dedupe from BBTools[65] using a threshold of 99.9% to create a database of unique genomes. A variant calling was performed using the program Snippy with default values using the strain NCTC 8325 as reference. The individual output files were concatenated in one table and the analysis and visualisation of variants for the genes *hsdR*, *hsdM* and *hsdS* were performed in R 3.4.

### Core gene SNP distance
Genomes were annotated using Prokka v1.14[66] and a core alignment Panaroo v1.5[67] in strict mode with Mafft aligner. The SNP distance for the strains from CC1 was determined using Panaroo core alignment as input for Snp-dists v0.7 with default parameters (Supplementary Tables 5 and 6).

### Population analysis
A Mash distances tree was created for all *S. aureus* complete genomes using the software MashTree v1.4.6 with default parameters[68].

### MLST and clonal complexes
Sequence typing was performed using the program mlst on all the complete *S. aureus* genomes. A table correlating ST and CCs for *S. aureus* was downloaded from pubMLST. Clonal complexes were assigned to each genome using the function match in R 3.4.

### MGE search
A phage and PICI BLAST search was performed using the sequence of the phage and PICI integrases. Additionally a plasmid BLAST search was carried out using the PLSDB database as query. A table containing the MGE content of the collection can be found in Supplementary Table 3.

### TDR identification
Recognition sequences of all the strains used in this study were determined using the REBASE database. Sequences of the *hsdS* genes were extracted and their target sequences were identified by using the Blast tool of REBASE[43]. A table showing the sequences recognised by the strains used in this study is provided in Supplementary Table 4.

### Complementation assays
*HsdR* was cloned in the vector pCN51. Complementation was assessed by performing transductions of the phage 80α on the strains carrying the plasmid pCN51(*hsdR*) or the empty vector and titering the transductant cells onto TSA chloramphenicol plates to select for 80α lysogens.

### Mutagenesis
Deletion of the R subunit (*hsdR*) was performed as previously described[68,69]. Briefly flanking regions of *hsdR* were cloned in the plasmid pMAD. The plasmid was introduced into RN4220 or the collection by electroporation or transduction and selected as dark blue colonies on TSA-Ery plates containing 80 µg/mL X-gal. For the first recombination step colonies carrying the plasmid pMAD(Δ*hsdR*) were grown overnight in TSB-Ery at 30 °C. The overnight cultures were diluted and plated onto pre-warmed TSA-Ery-Xgal plates and incubate at 42 °C for 48 h. Light blue colonies were selected and grown overnight in TSB with no antibiotic at 30 °C. The overnight cultures were diluted and plated onto pre-warmed TSA-Xgal plates and incubated at 42 °C for 48 h. White colonies were selected and streaked onto TSA

and TSA-Ery. The ones growing on TSA but not TSA-Ery were selected and the deletion was confirmed by PCR.

## Mixed population time-kill experiment

Overnight cultures of the R-deficient strain JP9179 and the R-functional strain JP9197 with a cadmium-resistance marker were normalised to an O.D.$_{600\,nm}$ of 1.4. These strains were combined in a 50:50 ratio in 3 ml TSB supplemented with 5 mM CaCl$_2$ and this mixed population was exposed to 150 µl of lysate (MOI of 1) from phage 80α carrying an erythromycin-resistance marker. The culture was incubated for 30 min (180 rpm, 37 °C) before erythromycin (10 µg/mL) was either added or not. The mixed population was incubated (180 rpm, 37 °C) for a further 8 h, and every 4 h, samples were extracted and serially diluted in 10-fold steps in phosphate-buffered saline (PBS). Dilutions were plated on TSA alone TSA containing cadmium chloride, TSA containing erythromycin, and TSA containing both cadmium chloride and erythromycin. Agar plates were incubated statically in air at 37 °C overnight to enumerate CFUs. The relative abundance of the R-functional strain JP9197 in the mixed population over time was calculated by dividing the number of CFUs on cadmium-supplemented TSA by the total number of CFUs on TSA alone.

## Statistical analysis

Shapiro–Wilk tests were performed to test for normality of the data. Welch t-tests (two-sided) were used to compare differences in efficiency of transfer between mutants and parental strains for three independent biological replicates of each ($n = 3$, Figs. 1, 3 and 4). When comparing against multiple timepoints (Fig. 6) an Anova test was performed for six biological replicates of each strain and condition ($n = 6$). For multiple test comparisons, a Bonferroni correction was used. A Poisson test was performed to test for significance on the number of loss-of-function mutations (Fig. S4).

## Reporting summary

Further information on research design is available in the Nature Portfolio Reporting Summary linked to this article.

# Data availability

The genomics data generated in this study have been deposited in the GenBank database under accession code BioProject ID PRJNA1177260. The experimental data generated in this study are provided in the Source data file. Source data are provided with this paper.

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

## Acknowledgements

We would like to thank Prof Luca G. Guardabassi for gifting us the strains used in this study as recipients. We would like to thank Lorrie Maccario for her assistance with library preparation and hybrid assemblies. This work was supported by grants MR/X020223/1, MR/M003876/1, MR/V000772/1 and MR/S00940X/1 from the Medical Research Council (UK), BB/V002376/1 and BB/V009583/1 from the Biotechnology and Biological Sciences Research Council (BBSRC, UK), EP/X026671/1 from the Engineering and Physical Sciences Research Council (EPSRC, UK), and ERC-2023-SyG Project 101118890—TalkingPhages to J.R.P., a research grant (VIL58733-Weaponizable satellites) from VILLUM FONDEN to R.I.-C, and grant BB/W014920/1 from the Biological Sciences Research Council, and the Biotechnology and Biological Sciences Research Council institute strategic grant BBS/E/D/20002173 to J.R.F.

## Author contributions

J.R.P. conceived the study; W.F., A.S. and R.I.C. conducted the experiments; W.F., J.G. performed the bioinformatics analyses; W.F., A.S., J.G., J.R.F. and J.R.P. analysed the data; W.F. made all figures, J.R.P. wrote the manuscript with inputs from all the authors.

## Competing interests

The authors declare no competing interests.

## Additional information

**Supplementary information** The online version contains Supplementary material available at

