## [Transparent Peer Review File · Nature Communications]

Immune-deficient bacteria serve as gateways to genetic exchange and microbial evolution

Corresponding Author: Professor José Penadés

Version 0:

Reviewer comments:

Reviewer #1

(Remarks to the Author)

This manuscript presents a very comprehensive study of gene transfer efficiency by several mechanisms (phage lysogenic conversion, generalized and lateral transduction, phage satellites and plasmid conjugation) between and within CCs (clonal complexes) in *S. aureus*. The authors show that barriers to transfer between CCs are for a large part driven by Type I restriction-modification systems. Furthermore, this makes the few immune-deficient strains a “gateway” to transfer within new CCs, because immune-deficient strains are only defective in the restriction subunit but still able to methylate the transferred DNA, making it protected against restriction within the new CC. Finally, a bioinformatics analysis shows that immune-deficient strains that can play this gateway role are relatively common across *S. aureus* CCs.

Most of my major comments below are about nuancing the conclusions derived from experimental data, and claims of novelty, but this is overall a great paper and I have no issue with any of the experimental results themselves.

1) Some of the claims of novelty / surprising results should be nuanced throughout the manuscript or better defended and put in context of the existing literature. I agree the specific phenomenon described where immune-deficient cells then transfer efficiently within CCs due to restriction only being deficient and not methylation is a novel aspect, to my knowledge, and to me a really interesting one, but the fact that permissive strains are permissive is not exactly surprising.

Variation in permissiveness towards HGT has been known for a while, together with its role in amplifying gene transfer among bacteria, e.g. doi.org/10.1093/genetics/162.4.1525.

The trade-off between positive and negative effects of carrying immune systems for bacteria has also been explored previously see <https://doi.org/10.1371/journal.pgen.1003844>, or <https://onlinelibrary.wiley.com/doi/full/10.1111/jeb.12291> for a theoretical analysis.

The major role of RM Type I in blocking HGT between *S. aureus* CCs was also already known (with numerous papers rightly cited here, but the authors could give more detail on what MGEs and HGT mechanisms were studied previously and how their results add to or are surprising in the context of this earlier literature).

2) In the paragraph starting L162, generalisation from transfer of a few MGEs to efficiency of mechanisms of HGT in general is unwarranted. The scale of work performed here is impressive, yet it is still focused on a few model MGEs. In particular for plasmid conjugation, the results are all dependent on conjugative efficiency of a single conjugative plasmid (as the other tested plasmid is mobilizable, so will ultimately depend on pGO1 plasmid conjugation). Conjugation efficiency is extremely variable across conjugative plasmids, not only across strains see <https://doi.org/10.1016/j.plasmid.2020.102489> so using a single conjugative plasmid that might be on the low end of that scale does not allow the authors to generalise to the quantitative effect of conjugation overall.

3) Some of the wording used to discuss the results implies that observed phenotypes, in particular the spread of genes within CCs, are adaptive. It is not always clear for which entity (the gene, the MGE, the host cell lineage, the CC, the species?). Words like “evolve to” (L66), “strategy” in Fig 4 and throughout, “to overcome this problem” L102 should be avoided, or if kept, it should be made much clearer at which level selection would be acting and how this would work (which is far from obvious and would probably require some kind of group selection). The results presented definitely show that there is variation in transfer, and the properties of immune-deficient strains will promote transfer within CCs, but they do not show (or require) that immune-deficient strains have evolved to this end.

Similarly, “there must be a fine balance” L386 is ambiguous – why must there be a balance, is that just to explain the results,

or does that imply this balance has been selected for?

4) The uncovered trade-off between phage susceptibility and gene acquisition, shown in the lab, does not necessarily explain persistence in nature, as claimed in the manuscript (abstract L31, results L291). The results here do suggest that the trade-off can or could explain persistence in nature, but conclusively proving that a pattern of natural variation is due to a given selective force would require stronger evidence like some measure of selection in natural environments or patterns of genome evolution.

5) The authors argue at several points in the manuscript (L35, L104, L242) that further dissemination within a CC requires the transferred genes to be beneficial. There is no more discussion or explanation of this point, and it is not clear to me why a benefit is required – pure infectious transmission of the MGE, or LT could be enough for further dissemination even without selection for the transferred genes.

Minor comments

Title – “immune-deficient” is a bit confusing outside of context. I am familiar with bacterial immune systems, and yet my first thought was wondering why I was sent a paper about bacterial interactions with the host immune system. Please consider rephrasing using bacterial immunity or defence systems to limit the confusion with animal immunity.

l63 confusion to clarify – antibiotic resistance is beneficial but virulence is harmful? It would be good to clarify anyway if the authors mean benefit or harm to the bacteria themselves or to humans.

l240 immune-competent immune-deficient?

L267 please provide more explanation on the statistical analysis or leave it all in the supplementary material – “p-value based on Poisson distribution” is not clear on its own as we have no idea what the Poisson distribution is about.

L275 a mechanism that “generates” these mutations - please reformulate, the point is about the mechanism being biased towards these specific mutations, not the mutations happening per se.

L345 where is the paradox? The authors next describe conflicting selective pressures acting on R-deficient strains, so a presence at low levels in the population is what would be expected.

Fig S1 and S2, the correlations or absence of correlations discussed would be easier to see directly with one HGT mechanism plotted against the other instead of separate plots – replicates of one can be plotted against the median or mean of the other.

Reviewer #2

(Remarks to the Author)

In this manuscript Figueroa et al. describe strains of *S. aureus* that are deficient in anti-MGE immune systems as a hub for acquiring MGEs to support *S. aureus* genome evolution under selective pressure. This largely resides in the observation that type-I RM systems are deficient in certain strains of *S. aureus* allowing for the permissive uptake of MGEs with a specific focus on phage lysogeny. These data support the idea that immune deficient strains, although at a fitness advantage under phage pressure, can benefit from more promiscuous DNA uptake in the face of the right selective pressures. Even more intriguing is the idea that once an MGE is embedded within an immune deficient strain, it can more readily expand these traits to other members of the same clonal complex. This latter point is the main focus of my most pressing comments for further consideration.

Major Comments

1. From Fig 4A, it makes sense that if the phages used originate from a donor (JP9179) that is immune proficient, then that same strain (JP9179), even though it has functional *hsdR*, would be more capable of being lysogenized, since this is self on self transmission. What would be more compelling would be to use these JP9179 originating phages on other CC1 immunocompetent strains to assess lysogeny, so strains closely related but not identical. This would be stronger evidence that once an MGE is seeded into a CC through an immune deficient strain, it is more likely to spread among others within that same complex that are immune competent.

2. Related to point 1. How distant do you have to be in terms of CC for the system to break? Clearly CC8 phages cannot be transferred to immunocompetent CC1 strains as shown in Fig 4B, but are more closely related CCs based on phylogeny more permissive to tolerate some level of lysogeny that is intermediate, even if they are immunocompetent?

3. Another important point has to do with the acquisition of phage by lysogeny and its benefit under selection (Fig 6). If there truly is a fitness tradeoff in harboring the phage, then in the absence of selection, newly lysogenized bacteria should be less fit in competition with their own isogenic strain background that lacks the phage, and/or when competing against an immune competent strain that also lacks the phage (i.e. JP9179 80a -vs- JP9197).

4. There are many strain designations to keep track of, especially for the lysogeny and competition experiments from Figs 3,4, 6. Perhaps each of these figures could have a panel with a schematic of how experiments were set up or what is being compared. I would have found such a visualization helpful in interpreting these data.

Minor Comments

1. Lines 67-69, on the activity of ICEs and IMEs, this sentence is difficult to understand and could be broken up into two separate sentences.
2. In the legend for Fig 1A there is no description of GT(80a) and pI258
3. Line 169-170, the statement that pC221 is not conjugative but mobilizable. For clarity I would reword more clearly to indicate that pC221 can be conjugated with the help of pGO1. As the sentence prior, line 169 indicates that pC221 can be mobilized by conjugation. I know its semantics between "conjugation" and "mobilization", and the end result is the same, but best to keep the terminology clear for those who have limited experience with MGE biology in bacteria.
4. I suggest taking the data from supplementary Fig 3 and incorporating that into Fig 1, since there are phages and elements that are critical additions to the analysis.
5. In Fig 3&4 it would be helpful to provide a label for recipient cells, so it is very clear that the strain designations on the X axis are the recipients.
6. There are few places where the text is denoted as immune-competent (see lines 240, and 243) but I think these should read immune-deficient. I suggest checking the document carefully.

REVIEWER COMMENTS

We thank the reviewers for their thorough, thoughtful, and constructive evaluation of our manuscript. We are grateful for the positive feedback and supportive comments, which we believe have helped us to significantly improve the clarity, balance, and contextualization of our work. We especially appreciate the reviewers' careful reading and insightful suggestions, which strengthened both the interpretation of our results and the presentation of the study. Their encouragement and detailed input were invaluable in refining the final version of the manuscript.

Reviewer #1 (Remarks to the Author):

This manuscript presents a very comprehensive study of gene transfer efficiency by several mechanisms (phage lysogenic conversion, generalized and lateral transduction, phage satellites and plasmid conjugation) between and within CCs (clonal complexes) in *S. aureus*. The authors show that barriers to transfer between CCs are for a large part driven by Type I restriction-modification systems. Furthermore, this makes the few immune-deficient strains a "gateway" to transfer within new CCs, because immune-deficient strains are only defective in the restriction subunit but still able to methylate the transferred DNA, making it protected against restriction within the new CC. Finally, a bioinformatics analysis shows that immune-deficient strains that can play this gateway role are relatively common across *S. aureus* CCs.

Most of my major comments below are about nuancing the conclusions derived from experimental data, and claims of novelty, but this is overall a great paper and I have no issue with any of the experimental results themselves.

We thank the reviewer for this positive and thoughtful evaluation of our work. We appreciate the constructive comments regarding interpretation and novelty, and have revised the manuscript to better nuance our conclusions and improve clarity.

1) Some of the claims of novelty / surprising results should be nuanced throughout the manuscript or better defended and put in context of the existing literature. I agree the specific phenomenon described where immune-deficient cells then transfer efficiently within CCs due to restriction only being deficient and not methylation is a novel aspect, to my knowledge, and to me a really interesting one, but the fact that permissive strains are permissive is not exactly surprising.

Variation in permissiveness towards HGT has been known for a while, together with its role in amplifying gene transfer among bacteria, e.g. doi.org/10.1093/genetics/162.4.1525.

The trade-off between positive and negative effects of carrying immune systems for bacteria has also been explored previously see <https://doi.org/10.1371/journal.pgen.1003844>, or <https://onlinelibrary.wiley.com/doi/full/10.1111/jeb.12291> for a theoretical analysis.

The major role of RM Type I in blocking HGT between *S. aureus* CCs was also already known (with numerous papers rightly cited here, but the authors could give more detail on what MGEs and HGT mechanisms were studied previously and how their results add to or are surprising in the context of this earlier literature).

We thank the reviewer for these insightful suggestions and for pointing out these foundational studies. We agree that the general variation in permissiveness and the trade-offs of immune systems have been previously described. Our intention was not to claim these broad concepts as novel, but rather to highlight a specific, underappreciated mechanism by which these trade-offs manifest in *S. aureus*.

We have revised the manuscript to nuance our conclusions and to better contextualise our findings within the existing literature, and cited the papers proposed by the reviewer (L362-377). Specifically, we now provide a more detailed explanation of how our results regarding Type I RM systems (specifically the role of restriction-deficient/methylation-competent intermediates) compared to previous studies.

2) In the paragraph starting L162, generalisation from transfer of a few MGEs to efficiency of mechanisms of HGT in general is unwarranted. The scale of work performed here is impressive, yet it is still focused on a few model MGEs. In particular for plasmid conjugation, the results are all dependent on conjugative efficiency of a single conjugative plasmid (as the other tested plasmid is mobilizable, so will ultimately depend on pGO1 plasmid conjugation). Conjugation efficiency is extremely variable across conjugative plasmids, not only across strains see <https://doi.org/10.1016/j.plasmid.2020.102489> so using a single conjugative plasmid that might be on the low end of that scale does not allow the authors to generalise to the quantitative effect of conjugation overall.

We thank the reviewer for their comment. To answer their concerns, we have:

1) Performed new conjugation experiments with a new plasmid - pWBG731. Previous studies have proposed that there are three main families of conjugative staphylococcal plasmids: SK41/pGO1-like plasmids, pWBG4 and pWBG749 families (<https://doi.org/10.1128/microbiolspec.gpp3-0030-2018>). The new plasmid we've incorporated to our study is related to the pWBG749 family (98% identity, 46% coverage). These new results show the same patterns as the other conjugation experiments. This has been added to the violin plots in Figure 1B, and supplementary figure FS3.

2) Changed the wording of our conclusions throughout the text, and added some lines to the discussion to state the limitations of our study (L318-319).

3) Some of the wording used to discuss the results implies that observed phenotypes, in particular the spread of genes within CCs, are adaptive. It is not always clear for which entity (the gene, the MGE, the host cell lineage, the CC, the species?). Words like "evolve to" (L66), "strategy" in Fig 4 and throughout, "to overcome this problem" L102 should be avoided, or if kept, it should be made much clearer at which level selection would be acting and how this would work (which is far from obvious and would probably require some kind of group selection). The results presented definitely show that there is variation in transfer, and the properties of immune-deficient strains will promote transfer within CCs, but they do not show (or require) that immune-deficient strains have evolved to this end.

Similarly, "there must be a fine balance" L386 is ambiguous – why must there be a balance, is that just to explain the results, or does that imply this balance has been selected for?

We appreciate the reviewer's caution regarding the interpretation of these evolutionary drivers. We agree that the increased gene transfer observed in immune-deficient strains may be a consequence of the loss of RM systems rather than an evolved "strategy" for HGT. We have now changed our wording throughout the text.

Regarding L386, by "there must be a fine balance", we do not mean to imply that the system has been selected to reach a specific goal of gene transfer. Instead, we are describing a selective constraint: lineages that fail to balance these two opposing pressures (phage killing vs. adaptive stagnation) are likely to be lost from the population. We have revised the text (L405-407) to clarify that this refers to the ecological pressures that dictate the survival and frequency of these different phenotypes within the CC.

4) The uncovered trade-off between phage susceptibility and gene acquisition, shown in the lab, does not necessarily explain persistence in nature, as claimed in the manuscript (abstract L31, results L291). The results here do suggest that the trade-off can or could explain persistence in nature, but conclusively proving that a pattern of natural variation is due to a given selective force would require stronger evidence like some measure of selection in natural environments or patterns of genome evolution.

We have modified this in the manuscript to reflect that this trade-off is a potential explanation to their existence in nature.

5) The authors argue at several points in the manuscript (L35, L104, L242) that further dissemination within a CC requires the transferred genes to be beneficial. There is no more

discussion or explanation of this point, and it is not clear to me why a benefit is required – pure infectious transmission of the MGE, or LT could be enough for further dissemination even without selection for the transferred genes.

The reviewer made an excellent point that dissemination of an MGE/gene does not strictly require a fitness benefit, it just needs to not be detrimental enough to be purged. We have revised the manuscript to clarify that transmission within a CC can be neutral and not necessarily driven by positive selection.

Minor comments

Title – “immune-deficient” is a bit confusing outside of context. I am familiar with bacterial immune systems, and yet my first thought was wondering why I was sent a paper about bacterial interactions with the host immune system. Please consider rephrasing using bacterial immunity or defence systems to limit the confusion with animal immunity.

We understand the reviewer’s comment. However, we feel like this term truly reflects the state of R-deficient mutants in the same sense as an immune-deficient person, whose immune system is compromised or entirely absent, making it hard to fight infections. And therefore we would like to keep the title as is.

l63 confusion to clarify – antibiotic resistance is beneficial but virulence is harmful? It would be good to clarify anyway if the authors mean benefit or harm to the bacteria themselves or to humans.

We have reworded this sentence.

l240 immune-competent immune-deficient?

We have double-checked the entire manuscript to ensure that the correct term was used.

L267 please provide more explanation on the statistical analysis or leave it all in the supplementary material – “p-value based on Poisson distribution” is not clear on its own as we have no idea what the Poisson distribution is about.

We have clarified this in the figure legend of the supplementary figure.

L275 a mechanism that “generates” these mutations - please reformulate, the point is about the mechanism being biased towards these specific mutations, not the mutations happening per se.

We have reworded this section accordingly.

L345 where is the paradox? The authors next describe conflicting selective pressures acting on R-deficient strains, so a presence at low levels in the population is what would be expected.

We thank the reviewer for this observation. We agree that “paradox” may overstate the phenomenon given that the conflicting selective pressures (phage predation vs. HGT) provide a clear mechanistic explanation for the persistence of these strains. We have revised this sentence to frame this as an “evolutionary trade-off” rather than a paradox. (L348-349).

Fig S1 and S2, the correlations or absence of correlations discussed would be easier to see directly with one HGT mechanism plotted against the other instead of separate plots – replicates of one can be plotted against the median or mean of the other.

We thank the reviewer for their suggestion. We have made new figures plotting in one axis one mechanism, and in the other axis, the other mechanism. We have also added a diagonal to represent when the efficiencies of both mechanisms are the same, and to indicate that any points above or below that line indicate that one of the mechanisms is more efficient than the other.

Reviewer #2 (Remarks to the Author):

In this manuscript Figueroa et al. describe strains of *S. aureus* that are deficient in anti-MGE immune systems as a hub for acquiring MGEs to support *S. aureus* genome evolution under selective pressure. This largely resides in the observation that type-I RM systems are deficient in certain strains of *S. aureus* allowing for the permissive uptake of MGEs with a specific focus on phage lysogeny. These data support the idea that immune deficient strains, although at a fitness advantage under phage pressure, can benefit from more promiscuous DNA uptake in the face of the right selective pressures. Even more intriguing is the idea that once an MGE is embedded within an immune deficient strain, it can more readily expand these traits to other members of the same clonal complex. This latter point is the main focus of my most pressing comments for further consideration.

Major Comments

1. From Fig 4A, it makes sense that if the phages used originate from a donor (JP9179) that is immune proficient, then that same strain (JP9179), even though it has functional *hsdR*, would be more capable of being lysogenized, since this is self on self transmission. What would be more compelling would be to use these JP9179 originating phages on other CC1 immunocompetent strains to assess lysogeny, so strains closely related but not identical. This would be stronger evidence that once an MGE is seeded into a CC through an immune deficient strain, it is more likely to spread among others within that same complex that are immune competent.

We thank the reviewer for their comment. The experiment suggested by the reviewer has been performed, but it seems to be a confusion with the strain names. One of the strains (the donor) is called JP9179, whereas the strain with a functional *HsdR* is JP9197 (same clonal complex, but completely different strain). We have added a comment in the figure legend to clarify that the strains are different.

2. Related to point 1. How distant do you have to be in terms of CC for the system to break? Clearly CC8 phages cannot be transferred to immunocompetent CC1 strains as shown in Fig 4B, but are more closely related CCs based on phylogeny more permissive to tolerate some level of lysogeny that is intermediate, even if they are immunocompetent?

We thank the reviewer for their comment. Based on our mash distance tree from Figure 5 (which compasses ~1k complete genomes), CC8 (in grey) and CC1 (in green) are very closely related (next to each other in the tree). Other studies (e.g. <https://doi.org/10.1371/journal.ppat.1012378>) have also shown that CC8 and CC1 are close relatives. Apart from CC1, the second closest based on our mash distance tree that are also part of our strain collection would be CC15 (JP980, JP9182, JP9183, JP9185) and CC5 (JP9187), which also block phage transfer. In terms of T1RM S-subunit alleles (supplementary table 4), CC8 shares MS2 with strains from ST1 (JP9179 and JP9181), and CC130 (JP9143, JP9172 and JP9173). In this case, we do see that these strains tend to be more permissive, which is in line with our conclusions.

3. Another important point has to do with the acquisition of phage by lysogeny and its benefit under selection (Fig 6). If there truly is a fitness tradeoff in harboring the phage, then in the absence of selection, newly lysogenized bacteria should be less fit in competition with their own isogenic strain background that lacks the phage, and/or when competing against an immune competent strain that also lacks the phage (i.e. JP9179 80a -vs- JP9197).

In this section, we refer as “tradeoff” to phage susceptibility. On one hand, phage susceptible stains (R-deficient strains) will be exposed to killing by phages and therefore phage-resistant strains would be preferred as they are not affected. But on the other hand, in the presence of certain selective pressure, such as antibiotics, only those R-deficient strains will be able to acquire beneficial MGEs that could help them survive antibiotic treatment.

4. There are many strain designations to keep track of, especially for the lysogeny and competition experiments from Figs 3,4, 6. Perhaps each of these figures could have a panel

with a schematic of how experiments were set up or what is being compared. I would have found such a visualization helpful in interpreting these data.

We thank the reviewer for their comment. We agree that adding schematics reflecting the experimental design makes the figures easier to understand. We have added new panels to Figures 3, 4 and 6, and modified the figure legends accordingly.

Minor Comments

1. Lines 67-69, on the activity of ICEs and IMEs, this sentence is difficult to understand and could be broken up into two separate sentences.

We have modified this paragraph to make it easier to understand.

2. In the legend for Fig 1A there is no description of GT(80a) and pI258

We thank the reviewer for the observation, we have modified the figure legend to include those descriptions.

3. Line 169-170, the statement that pC221 is not conjugative but mobilizable. For clarity I would reword more clearly to indicate that pC221 can be conjugated with the help of pGO1. As the sentence prior, line 169 indicates that pC221 can be mobilized by conjugation. I know its semantics between “conjugation” and “mobilization”, and the end result is the same, but best to keep the terminology clear for those who have limited experience with MGE biology in bacteria.

We thank the reviewer for bringing this to our attention. We have reworded the sentence to make it clear that pC221 needs pGO1 to be mobilised by conjugation.

4. I suggest taking the data from supplementary Fig 3 and incorporating that into Fig 1, since there are phages and elements that are critical additions to the analysis.

We appreciate the reviewer’s comment. In Figure 1B, we have already incorporated all the data from supplementary figure 3. Panel 1A only shows some representative MGEs, however, panel 1B includes the data of all the MGEs tested. L175-176 clarifies this in the text.

5. In Fig 3&4 it would be helpful to provide a label for recipient cells, so it is very clear that the strain designations on the X axis are the recipients.

We have followed the reviewer’s suggestion and modified the labels in the x-axis to make it clear that those strains are the recipients.

6. There are few places where the text is denoted as immune-competent (see lines 240, and 243) but I think these should read immune-deficient. I suggest checking the document carefully.

We thank the reviewer for the observation, we have corrected this throughout the document.